# Three-Dimensional Model of Soil Water and Heat Transfer in Orchard Root Zone under Water Storage Pit Irrigation

**Yuanyuan Su** [1], **Xianghong Guo** [1,2,*], **Tao Lei** [1], **Lijian Zheng** [1], **Juanjuan Ma** [1], **Xihuan Sun** [1], **Linru Hao** [1] **and Feipeng Hu** [1]

1 College of Water Resource Science and Engineering, Taiyuan University of Technology, Taiyuan 030024, China; suyuanyuan0541@link.tyut.edu.cn (Y.S.); lcsyt@126.com (T.L.); zhenglijian@tyut.edu.cn (L.Z.); mjjsxty@126.com (J.M.); sunxihuan@tyut.edu.cn (X.S.); haolinru0756@link.tyut.edu.cn (L.H.); hufeipeng0534@link.tyut.edu.cn (F.H.)

2 State Key Laboratory of Simulation and Regulation of Water Cycle in River Basin, China Institute of Water Resources and Hydropower Research, Beijing 100038, China

\* Correspondence: guoxianghong@tyut.edu.cn

**Abstract:** To reveal the water and heat transfer characteristics of the orchard soil under water storage pit irrigation and to regulate the distribution of soil water and heat for improving apple quality and increasing yield, a 3D soil water and heat transfer model of orchards under water storage pit irrigation was established. The model not only considered the influences of root water uptake, precipitation, evaporation, and irrigation, but also simulated the infiltration process of the variable water head in the pit according to the principle of mass conservation and introduced the pit coefficient to simulate the difference in radiation in the pit to describe the influence of the pit on the model. Verify and analyze the simulation results. Results showed that the variation trend of simulated soil moisture and heat was consistent with that of measured data. The mean absolute percentage error, root mean square error, and mean absolute deviation were 3.23%, 0.9460, and 0.6984 for soil temperature and 10.05%, 0.0269, and 0.0214 for water content after irrigation, respectively. The simulation results have high accuracy and show that the soil moisture content centers on the pit with an ellipsoid distribution and tends to be uniform over time. The soil temperature was higher in the 4–5 cm area near the soil surface and the wall of pit, and it remarkably changed with time. The intraday variation of soil temperature was mainly affected by atmospheric temperature, but a certain lag was observed compared with the change of atmospheric temperature. With the increase of the irrigation amount, the distribution range of soil moisture content and water high value area increased, while the average and maximum soil temperature decreased. With the increase of irrigation water temperature to 18–24 h after irrigation, the soil temperature in the ellipsoidal area around the pit remarkably increased. The model established in this paper can be used to simulate the hydrothermal status of the soil in the field under water storage pit irrigation. The results prove that the water storage pit irrigation can effectively improve the hydrothermal status of the middle-deep soil and promote the root system of fruit trees to absorb water.

**Keywords:** water storage pit irrigation; hydrothermal coupling; finite element method; numerical simulation

## 1. Introduction

Soil water and heat are essential elements for crop growth and play a vital role in this process. Soil moisture is the main source of water absorption by plants. It is an essential medium for chemical, biological, and physical processes in soil, it affects the absorption of plant nutrients, soil fertility, soil temperature, and ventilation, and it plays an important role in plant yield and quality. As an important factor in the ecological environment, soil temperature and its changes affect the microbial activity in the soil and

some physical and chemical characteristics of soil aqueous solutions [1], thus affecting the water movement, soil fertility, and crop growth.

Water movement and heat transfer in soil are not independent but mutually influencing processes [2,3], and the beneficial coupling of hydrothermal can promote water and nutrient absorption by roots, thereby promoting plant growth and improving crop quality [4]. Many common factors affect and regulate soil moisture and heat, including meteorology [5,6], soil structure and properties [7], farmland management regime (e.g., mulching) [8], and irrigation. Irrigation is commonly used to regulate farmland water and heat. Different irrigation methods involve different soil water distribution characteristics and different effects on soil temperature. Lv, et al. (2012) [9] conducted field experiments and found that the surface temperature of the sprinkler irrigation was low, and the profile distribution of soil temperature, except for the surface layer, could be approximated as an exponential function distribution; moreover, the surface border irrigation profile distribution of soil temperature indicated an exponential distribution, high-frequency irrigation resulted in low surface soil temperature under drip irrigation, the highest temperature was observed at approximately 20 cm deep in the soil, and the profile temperature distribution presented an "S" shape characteristic. Ding, et al. (2019) [10] found that compared with furrow irrigation, drip irrigation had a lower cooling amplitude after irrigation, soil temperature recovered quickly, and the depth of the soil layer affected by irrigation was shallow. Li, et al. (2014) [11] found that compared with conventional furrow irrigation, alternate furrow irrigation was conducive to the lateral infiltration of soil moisture, increased the soil temperature in the root zone, and resulted in uniform distribution. Different irrigation methods have different effects on soil water and heat conditions. Water storage pit irrigation is a water-saving irrigation method, which can solve the problems of drought and soil erosion and is suitable for fruit forest irrigation in hilly areas. In the field, several small water storage pits are evenly arranged under the canopy along the circumference of half of the canopy radius. During irrigation, water can be transported and injected into the pits through pipeline or field channel, and then infiltrate into the soil in the root zone along the pit wall. The field layout is shown in Figure 1. This irrigation method has the advantages of water saving, water retention, drought resistance, and induced deep rooting [12]. The biggest difference between water storage pit irrigation and other irrigation methods is the layout of water storage pit, which changes the infiltration interface of orchard irrigation water (from the surface to the pit wall), resulting in high middle-deep soil moisture content in the root area of the orchard and low value in the surface. This condition can effectively reduce the evaporation between trees and improve water use efficiency. The pit wall of the water storage pit is also a free surface, which increases the exchange area between the middle-deep soil and the atmospheric temperature and changes the temperature distribution of the middle-deep soil [13]. The soil hydrothermal status under the water storage pit irrigation is different from other irrigation methods. The current research on the soil water and heat under water storage pit irrigation is not comprehensive. Most of the researches focus on the study of soil water movement under water storage pit irrigation [14–17], and there are fewer simulated studies of soil temperature. Some scholars have used BP neural networks to predict soil temperature under water storage pit irrigation [18,19]. The model of the water-heat movement mechanism under the water storage pit irrigation is even less. The 2D mathematical model compiled by Ren (2018) [4] under the condition of single-pit irrigation for water storage only considers the indoor 2D conditions. Wang (2018) [20] used COMSOL software to establish a 3D soil water-heat coupling model for orchards during freeze-thaw period under water storage pit irrigation, but the model did not consider the influence of root water absorption and vegetation, and was not suitable for other climatic conditions. A 3D hydrothermal coupling model for the non-freeze-thaw period under water storage pit irrigation needs to be established. Hence, the soil hydrothermal status under the water storage pit irrigation should be explored, the beneficial coupling of soil water and heat should be promoted, and the fruit yield and quality should be improved.

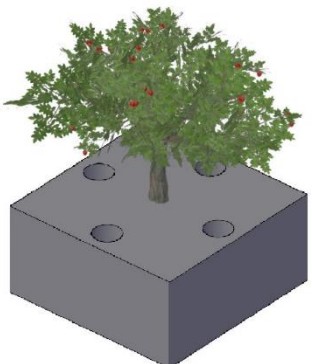

**Figure 1.** Field layout.

In recent years, numerical simulation has become an important means to study soil hydrothermal properties. Researchers had successively used soil water content and soil temperature, soil matrix potential and temperature, soil matrix potential gradient and temperature, and total soil water mass and total soil enthalpy as unknowns; they have also constructed soil water-gas-heat [21], water-vapor-heat [22–25], water-heat-solute [26,27], and soil water and heat transport coupling models [28]. Coupling model not only considers soil module to construct soil hydrothermal coupling model [29–31] but also considers the energy budget of vegetation to establish soil-plant-atmosphere transfer model, such as simple soil-plant-atmosphere transfer model (SiSPAT) [32]. In addition, the soil hydrothermal model and crop model were combined to develop integrated soil-crop model, such as the WHCNS model [33,34], and improve it [35]. They used the model to analyze the influence of different factors on the model, including convective water vapor flux [36], soil moisture evaporation [29], irrigation water temperature [37], solar radiation, air temperature, initial soil temperature and initial soil water content [30], temperature gradient, and root water absorption [31]. However, the soil hydrothermal model is still dominated by 1D models, and the development of 2D and 3D models is still relatively slow. HYDRUS-2D/3D is constructed by Šimůnek et al. (2020) [38]; it can be used to simulate water, heat, and solute movement in 2D and 3D variably saturated porous media, and the software has been applied and evaluated by researchers [39–41]. The simulation performance of HYDRUS depends heavily on the choice of different surface boundaries [40]; the software is a commercial closed-source software that is limited in terms of dealing with complex boundary conditions. The soil moisture and heat movement of water storage pit irrigation is a typical three-dimensional problem, and the soil hydrothermal boundary conditions of the pit wall are complex, making it hard to apply the above model directly. Therefore, the soil hydrothermal coupling model of water storage pit irrigation under complex field conditions should be further studied and established.

This study aims to (1) establish a three-dimensional soil hydrothermal coupling model for water storage pit irrigation under complex field conditions based on consideration of the influence of canopy and water storage pit on soil water and heat, (2) verify the correctness of the soil water-heat coupling model of water storage pit irrigation by using field data, and (3) analyze the spatial distribution characteristics and inter-day and intra-day dynamic changes of soil water and heat in the orchard under water storage pit irrigation.

## 2. Materials and Methods

### 2.1. Study Area

The field study was conducted at the Fruit Research Institute of Shanxi Academy of Agricultural Sciences (112°32′ E, 37°23′ N), the location of study area is shown in Figure 2. The region is located in the southwest of Taigu County, Shanxi Province, China. The region has a typical temperate continental climate with a long-term annual mean air temperature and precipitation of 9.8 °C and approximately 460 mm, respectively. The average altitude is 781.9 m, the frost-free period is 175 days. The soil type in the test area is mainly silt loam,

the average dry bulk density was 1.47 g·cm$^{-3}$, and the soil basic physical parameters in the study area were shown in Table 1. The irrigation water source is groundwater.

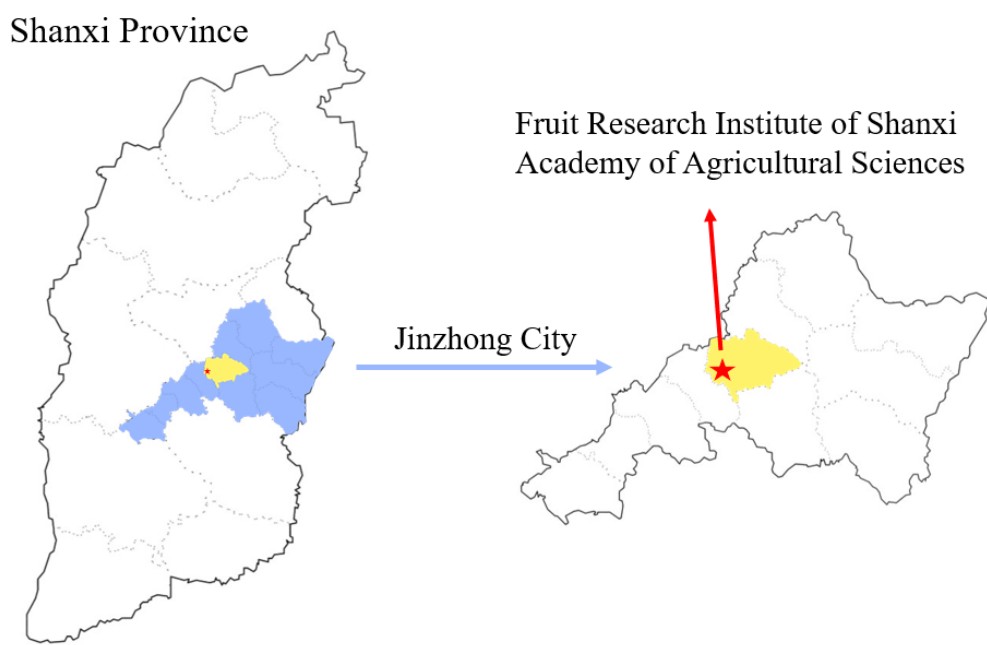

**Figure 2.** The location of study area.

**Table 1.** Soil physical parameters.

| Soil Depth/cm | $\theta_f$/cm$^3$·cm$^{-3}$ | $\theta_s$/cm$^3$·cm$^{-3}$ | Dry Bulk Density/g·cm$^{-3}$ | Soil Texture |
|---|---|---|---|---|
| 0–40 | 0.30 | 0.51 | 1.49 | silt loam |
| 40–70 | 0.28 | 0.52 | 1.44 | silt loam |
| 70–120 | 0.29 | 0.49 | 1.56 | silt loam |
| 120–170 | 0.32 | 0.50 | 1.51 | loam |
| 170–200 | 0.30 | 0.52 | 1.45 | loam |

*2.2. Experimental Design and Test Items*

2.2.1. Experimental Design

The fruit trees for test were dwarf anvil densely planted red Fuji apple trees with a row and plant spacing of 4 and 2 m, respectively. The test layout is shown in Figure 1. With the fruit tree as the center, four water storage pits with a diameter of 30 cm were excavated at a distance of 60 cm from the fruit trees, and they were evenly distributed around the fruit trees. The pit depth was 40 cm, the bottom of the pit was impermeable, and the irrigation water penetrated into the soil of the root area through the pit wall during the test.

The experiment was conducted from 24 June to 11 July 2019, the rainfall during the test period is 3.4 mm. The soil water content before irrigation was measured and recorded on 24 June, irrigation was carried out on the same day at 11:00, and the soil moisture content and soil temperature were measured again on 25 June and 11 July (1 and 17 days after irrigation).

2.2.2. Test Items

(1) Soil moisture content: The soil volumetric moisture content was measured using the TRIME-PICO IPH measuring system (TDR). The layout of measuring points is shown in Figure 3. Six measurement points were set for water content, namely, points 1, 2, 3, 4, 5, and 6. The measuring depth was 200 cm, and the vertical measurement spacing was 20 cm.

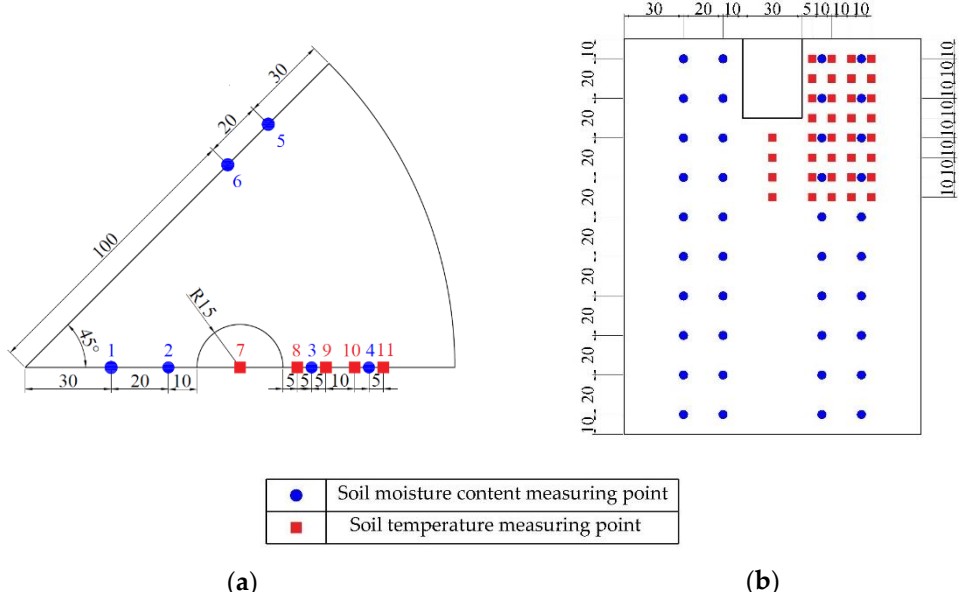

(a)                                  (b)

**Figure 3.** Distribution of soil moisture content and temperature measurement points. (**a**) Horizontal distribution; (**b**) Vertical distribution.

(2) Soil temperature: The soil temperature was measured using the multi-channel soil temperature tester, and the data were recorded every 30 min during the observation period. The arrangement of the temperature probe is shown in Figure 3. The water storage pit was set as the measurement center, and the soil temperature around the pit was monitored. Five temperature measurement points were arranged, namely, points 7, 8, 9, 10, and 11. The vertical arrangement of temperature probes is as follows: point 7 is arranged every 10 cm below the irrigation pit, and four probes were arranged. The other points are arranged every 10 cm from the ground surface, and eight probes were arranged at each point.

(3) Leaf area index (LAI): The leaf area index was measured using the LAI-2200 vegetation canopy analyzer, and data were collected once during the trial.

(4) Meteorological data: The meteorological data required for the experiment, including solar radiation, precipitation, atmospheric temperature, relative humidity, and wind speed, were collected from the ADCON wireless automatic meteorological station. $ET_0$ and atmospheric temperature changes during the experiment are shown in Figure 4. The rainfall and irrigation during the experiment are shown in Table 2.

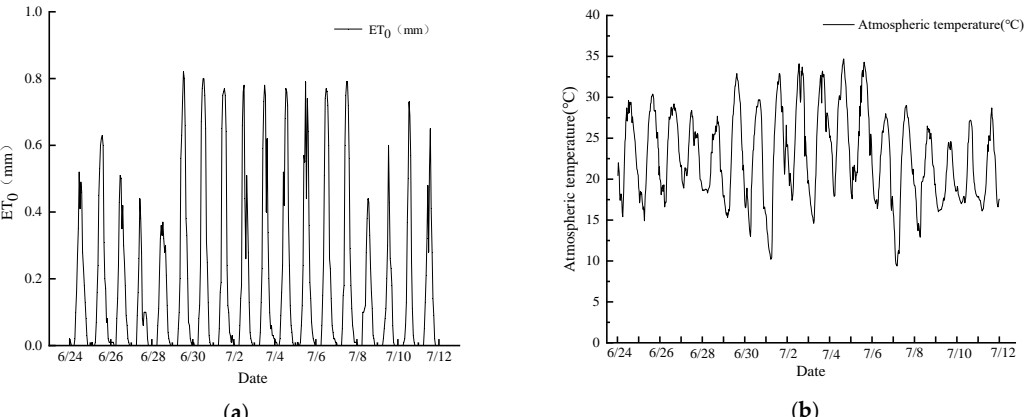

(a)                                  (b)

**Figure 4.** Variation of $ET_0$ and atmospheric temperature during the experiment. (**a**) Variation of $ET_0$ during the experiment; (**b**) Variation of atmospheric temperature during the experiment.

**Table 2.** Rainfall and irrigation during the experiment.

| Date | Rainfall/mm | Irrigation/L |
|---|---|---|
| 24 June | | 300 |
| 27 June | 0.6 | |
| 28 June | 0.2 | |
| 3 July | 0.6 | |
| 9 July | 2.0 | |

*2.3. Establishment of a Three-Dimensional Model of Soil Water and Heat Transfer in Orchard under Water Storage Pit Irrigation*

2.3.1. Determination of Simulation Area

The field layout of water storage pit irrigation is shown in Figure 5. During irrigation, water enters the water storage pit and infiltrates evenly from the pit wall to the surrounding area. With the passage of time, the infiltration area continues to expand outward and finally intersects in the middle of the two water storage pits, forming a zero-flux boundary. Moreover, forward water movement is no longer observed, and it moves vertically downward instead. According to the symmetry of the water distribution, the area from the pit center to the zero-flux surface, that is, the area between the ADLs, was selected for simulation calculation. The computational region is shown in Figure 6.

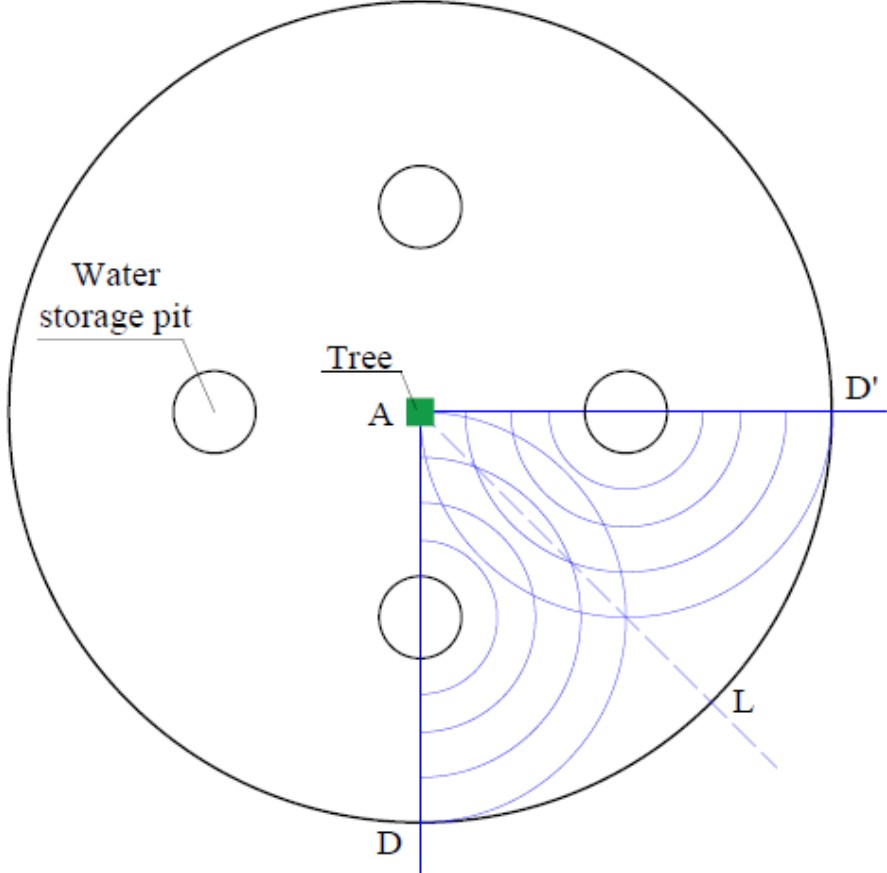

**Figure 5.** Field layout of water storage pit irrigation.

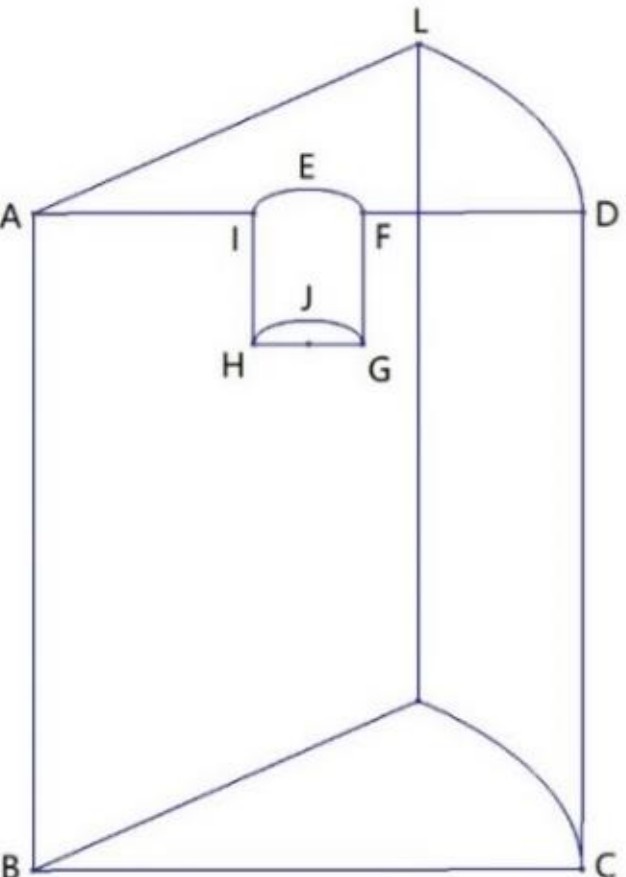

**Figure 6.** Computational region of water storage pit irrigation.

### 2.3.2. Governing Equation

(1) Governing equations of soil moisture movement

Each layer of soil was assumed to be homogeneous and isotropic; the infiltration water flow is a continuous medium and incompressible, and the soil skeleton is not deformed during soil moisture movement. The effect of temperature on soil water movement is minimal [37]. Therefore, ignoring the effect of temperature on water movement, the three-dimensional equation of soil water movement that considers root water uptake is expressed in Equation (1) [17] as follows:

$$\frac{\partial \theta}{\partial t} = \frac{\partial}{\partial x}\left[K(h)\frac{\partial h}{\partial x}\right] + \frac{\partial}{\partial y}\left[K(h)\frac{\partial h}{\partial y}\right] + \frac{\partial}{\partial z}\left[K(h)\frac{\partial h}{\partial z}\right] + \frac{\partial K(h)}{\partial z} - S \tag{1}$$

where $h$ is the soil water matrix potential represented by negative-pressure head in cm, $\theta$ is the soil volumetric moisture content in $cm^3 \cdot cm^{-3}$, $t$ is time in min, $x$, $y$, and $z$ are the space coordinates, where $z$ up is positive, in cm, $K(h)$ is the unsaturated hydraulic conductivity in cm/min, and $S$ is the root water uptake rate in L/min.

The Van Genuchten model [42] (VG equation) proposed by Van Genuchten in 1980 was used to represent the soil hydraulic properties, as follows:

$$\theta = \theta_r + \frac{\theta_s - \theta_r}{\left[1 + |\alpha h|^n\right]^m} \tag{2}$$

where $\theta_s$ is the saturated moisture content in $cm^3 \cdot cm^{-3}$, $\theta_r$ is the residual moisture content in $cm^3 \cdot cm^3$, $\alpha$, $m$, and $n$ are the shape parameters of soil water characteristic curve, and $m = 1 - \frac{1}{n}, (n > 0)$.

The unsaturated hydraulic conductivity can be expressed as follows:

$$K(h) = \begin{cases} K_s S_e^{\frac{1}{2}} \left[ 1 - \left( 1 - S_e^{\frac{1}{m}} \right)^m \right]^2 & , h < 0 \\ K_s & , h \geq 0 \end{cases} \tag{3}$$

where $K_s$ is the saturated hydraulic conductivity in cm/min, and $S_e = \frac{\theta - \theta_r}{\theta_s - \theta_r}$.

The root water uptake model [43] is as follows:

$$S(x, y, z, t) = \gamma(h) S_{max}(x, y, z, t) \tag{4}$$

where $S(x, y, z, t)$ is the actual root water uptake rate in L/min, $S_{max}(x, y, z, t)$ is the maximum root water uptake rate in L/min, and $\gamma(h)$ is the water stress coefficient.

$$\gamma(h) = \begin{cases} \frac{h_0 - h}{h_0 - h_1} & , h_1 \leq h \leq h_0 \\ 1 & , h_2 \leq h \leq h_1 \\ \frac{h - h_3}{h_2 - h_3} & , h_3 \leq h \leq h_2 \\ 0 & , h \leq h_3 \end{cases} \tag{5}$$

where $h_0$, $h_1$, $h_2$, and $h_3$ are the matrix potential heads corresponding to $\theta_s$, $80\%\theta_f$, $60\%\theta_f$, and $\theta_w$.

$S_{max}(x, y, z, t)$ can be expressed using the expression of Vrugt et al. (2001) [44] as follows:

$$S_{max(x,y,z)} = \frac{A_{xy}\beta(x, y, z)T_{pot}}{\int_0^{X_m} \int_0^{Y_m} \int_0^{Z_m} \beta(x, y, z) dx dy dz} \tag{6}$$

where $X_m$, $Y_m$, and $Z_m$ are the maximum extension depths (cm) of root in the $x$, $y$ and $z$ directions, respectively, $A_{xy}$ is the area of computational region on the surface in cm$^2$, and $\beta(x, y, z)$ is the shape factor that describes the spatial distribution of root water uptake potential.

$$\beta(x, y, z) = \left( 1 - \frac{x}{X_m} \right) \left( 1 - \frac{y}{Y_m} \right) \left( 1 - \frac{z}{Z_m} \right) e^{-\left[ \frac{p_x}{X_m} |x^* - x| + \frac{p_y}{Y_m} |y^* - y| + \frac{p_z}{Z_m} |z^* - z| \right]} \tag{7}$$

where $p_x$, $p_y$, $p_z$, $x^*$, $y^*$, and $z^*$ are the fitting parameters.

$T_{pot}$ is the potential transpiration intensity in cm/min as follows:

$$T_{pot} = K_c ET_0 \tag{8}$$

where $K_c$ is the crop coefficient obtained from the *Crop evapotranspiration: Guidelines for computing crop water requirements* published by FAO [45], and $ET_0$ (cm/min) is the reference crop evapotranspiration collected from the meteorological station.

(2)　Governing equations for soil heat transfer

Considering the heat transfer generated by heat conduction and convection and ignoring the influence of heat radiation, the three-dimensional soil heat transfer governing equation is obtained [46].

$$c_v \frac{\partial T}{\partial t} = \frac{\partial}{\partial x} \left( K_h \frac{\partial T}{\partial x} \right) + \frac{\partial}{\partial y} \left( K_h \frac{\partial T}{\partial y} \right) + \frac{\partial}{\partial z} \left( K_h \frac{\partial T}{\partial z} \right) - c_w q_x \frac{\partial T}{\partial x} - c_w q_y \frac{\partial T}{\partial y} - c_w q_z \frac{\partial T}{\partial z} \tag{9}$$

where $T$ is the soil temperature in °C, $c_v$ is the soil heat capacity in J/(cm$^3$·°C), $K_h$ is the thermal conductivity in J/(cm·min·°C), and $c_w$ is the specific heat capacity of water in J/(cm$^3$·°C).

The value of $C_v$ is related to soil composition and moisture content. Ignoring the low content of organic matter in the soil, the soil heat capacity can be expressed by the water content [47] as follows:

$$C_v = 1.926(1 - \theta_s) + 4.184\theta \tag{10}$$

$K_h$ is related to moisture content, and the result of Chung et al. (1987) [48] was used for model simulation.

$$K_h = b_1 + b_2\theta + b_3\theta^{0.5} \tag{11}$$

where $K_h$ is the thermal conductivity in W/(m·°C), and $b_1$, $b_2$, and $b_3$ are the regression coefficients.

2.3.3. Definite Conditions
Initial Conditions

$$h(x, y, z, t) = h_0(x, y, z) \quad t = 0 \tag{12}$$

$$T(x, y, z, t) = T_0(x, y, z) \quad t = 0 \tag{13}$$

where $h_0(x, y, z)$ is the negative pressure head at the initial moment, and $T_0(x, y, z)$ is the initial soil temperature at each point.

Boundary Conditions

To surface boundary (ALDFEI, refer to Figure 6), during precipitation, the change of surface soil moisture boundary conditions is related to precipitation intensity $P$ and water infiltration intensity $i$. When $P < i$, the Neumann boundary condition is considered; when $P > i$, it forms a standing water condition known as the Dirichlet boundary condition.

$$-K(h)\left(\frac{\partial h}{\partial z} + 1\right) = P \quad t > 0, P < i \tag{14}$$

$$h(x, y, z) = h_s \quad t > 0, P > i, z = z_u \tag{15}$$

where $P$ is the net precipitation intensity in cm/min, $i$ is the infiltration intensity in cm/min, $z_u$ is the $z$ coordinate value of the surface boundary, and $h_s$ is the surface ponding depth in cm.

During evaporation, the surface soil moisture boundary is the Neumann boundary condition; when the ground becomes dry, the boundary is converted to the Dirichlet boundary condition.

$$-K(h)\left(\frac{\partial h}{\partial z} + 1\right) = e_s \quad t > 0, h(x, y, z_u) > h_d \tag{16}$$

$$h(x, y, z) = h_d \quad t > 0, h(x, y, z_u) \leq h_d, z = z_u \tag{17}$$

where $e_s$ is the surface evaporation intensity in mm/d, and $h_d$ is the suction head when the surface is dry expressed in cm.

Considering that the surface temperature is greatly affected by solar radiation, the surface soil temperature boundary is regarded as the Neumann boundary condition.

$$-K_h\frac{\partial T}{\partial z} = g(t) \quad z = z_u, \, t > 0 \tag{18}$$

where $g(t)$ is the known heat flux density function in J/(cm$^2$·min), which is calculated according to the energy balance equation.

To pit wall boundary (IEFGJH, refer to Figure 6), during irrigation, the soil moisture boundary is the Dirichlet boundary condition. When the infiltration in the pit is completed, the pit wall soil moisture boundary is regarded as the Neumann boundary condition.

$$h(x, y, z) = h_i \quad (x, y, z)\Gamma_2, t > 0 \tag{19}$$

$$-K(h)\left(\frac{\partial h}{\partial x}n_x + \frac{\partial h}{\partial y}n_y\right) = e_p \qquad (x,y,z)\Gamma_2, t > 0 \qquad (20)$$

where $\Gamma_2$ is the boundary area of the pit wall, $h_i$ is the suction head acting on each point of the pit wall expressed in cm, and $e_p$ is the evaporation intensity of pit wall expressed in mm/d.

During irrigation, the soil temperature boundary is Dirichlet boundary condition, and the soil temperature is considered to be the same as the water temperature. When the water infiltration in the pit is finished, the soil temperature boundary is converted to the Neumann boundary condition.

$$T(x,y,z,) = T_w \quad (x,y,z)\Gamma_2, t > 0 \qquad (21)$$

$$-K_h\left(\frac{\partial T}{\partial x}n_x + \frac{\partial T}{\partial y}n_y\right) = g_k(x,y,z,t) \qquad (x,y,z)\Gamma_2, t > 0 \qquad (22)$$

where $T_w$ is the temperature of irrigation water in °C, and $g_k(x,y,z,t)$ is the heat flux density function in the pit in $J/(cm^2 \cdot min)$.

To pit bottom boundary (HJG, refer to Figure 6), the pit bottom boundary is impermeable and the water flux is 0:

$$-K(h)\left(\frac{\partial h}{\partial z} + 1\right) = 0 \quad t > 0 \qquad (23)$$

Although no water movement is observed at the pit bottom, heat transfer still occurred. During irrigation, the soil temperature boundary is the Dirichlet boundary condition, and the soil temperature is considered to be the same as the water temperature. After water infiltration in the pit was completed, the soil temperature boundary is converted to the Neumann boundary condition.

$$T(x,y,z,) = T_w \quad z = 0, \ 0 < t < t_1 \qquad (24)$$

$$-K_h\frac{\partial T}{\partial z} = g_k(t) \quad z = 0, \ t > t_1 \qquad (25)$$

where $t_1$ is the end time of water infiltration.

To lower boundary (BCK, refer to Figure 6), considering the deep calculation depth and the large burial depth of groundwater, the water movement in this region is not affected, the soil temperature slightly changed compared with surface temperature, and the lower boundary can be regarded as Dirichlet boundary condition.

$$h(x,y,z,t) = h_0(x,y,z) \quad z = z_d, \ t > 0 \qquad (26)$$

$$T(x,y,z,t) = T_0(x,y,z) \quad z = z_d, \ t > 0 \qquad (27)$$

where $z_d$ is the $z$ coordinate value corresponding to the boundary of the pit bottom.

To boundary ABCD (Refer to Figure 6), boundary ABCD is symmetric, water flux and soil heat flux are 0.

$$-K(h)\frac{\partial h}{\partial y} = 0 \quad (x,y,z)\Gamma_3, t > 0 \qquad (28)$$

$$-K_h\frac{\partial T}{\partial y} = 0 \quad (x,y,z)\Gamma_3, t > 0 \qquad (29)$$

where $\Gamma_3$ is the area of boundary ABCD.

To boundary ABKL (Refer to Figure 6), boundary ABKL is symmetric and water flux and soil heat flux are 0.

$$-K(h)\left(\frac{\partial h}{\partial x}n_x + \frac{\partial h}{\partial y}n_y\right) = 0 \quad (x,y,z)\Gamma_4, \ t > 0 \qquad (30)$$

$$-K_h\left(\frac{\partial T}{\partial x}n_x + \frac{\partial T}{\partial y}n_y\right) = 0 \quad (x,y,z)\Gamma_4 \,, \, t > 0 \tag{31}$$

where $\Gamma_4$ is the area of boundary ABKL.

To boundary LKCD (Refer to Figure 6), boundary LKCD can be regarded as Dirichlet boundary condition because of the large calculation.

$$h(x,y,z,t) = h_0(x,y,z) \quad (x,y,z)\Gamma_5 \,, \, t > 0 \tag{32}$$

$$T(x,y,z,t) = T_0(x,y,z) \quad (x,y,z)\Gamma_5 \,, \, t > 0 \tag{33}$$

where $\Gamma_5$ is the area of boundary LKCD.

2.3.4. Model Solving

At present, for the three-dimensional soil hydrothermal motion model, finite difference [30], finite volume [14], and finite element method [41] are generally used for solving. Among these methods, finite element method has good boundary conditions adaptability and is convenient for programming general programs. Therefore, Galerkin finite element method was used in this paper to solve the model. See Supplementary Materials for more details.

Calculation of Surface Soil Heat Flux

Soil heat flux was calculated according to the soil surface energy balance equation as follows:

$$g(t) = R_s - H - \lambda E_s \tag{34}$$

where $R_s$ is the net radiation on the soil surface in $\text{J}/(\text{cm}^2{\cdot}\text{min})$, $H$ is the sensible heat flux in $\text{J}/(\text{cm}^2{\cdot}\text{min})$, and $\lambda E_s$ is the latent heat of vaporization in $\text{J}/(\text{cm}^2{\cdot}\text{min})$.

Surface net radiation is an important part of the heat balance and simulation of soil temperature. Under complex field conditions, the influence of crop canopy in the simulation of radiation status should be considered. Beer' Law [49] was used in this paper to reflect the influence of complex canopy structure on the net radiation of soil surface.

$$R_s = R_n \exp\left(-\frac{C * LAI}{cos\theta}\right) \tag{35}$$

$$R_n = R_v + R_s \tag{36}$$

where $R_n$ is the total net radiation in $\text{J}/(\text{cm}^2{\cdot}\text{min})$, $LAI$ is the leaf area index in $\text{cm}^2{\cdot}\text{cm}^{-2}$; $C$ is the extinction coefficient of the vegetation canopy, which depends on the leaf angle distribution of the canopy elements with values ranging from 0.3 to 0.7 ($C = 0.5$ for the canopy with spherical leaf angle distribution) [50]; $\theta$ is the solar zenith angle in *rad*, and $R_v$ is the net radiation of vegetation canopy in $\text{J}/(\text{cm}^2{\cdot}\text{min})$.

$H$ was determined by the following equation according to the study of Campbell [51]:

$$H = -0.006\frac{\rho c_p(T_s - T_a)}{\gamma_H} \tag{37}$$

$$\gamma_{\text{H}} = \frac{1}{\kappa u_*}\left[\ln\left(\frac{z_{ref} - d_0 + z_h}{z_h}\right) + \psi_h\right] \tag{38}$$

$$u_* = u\kappa\left[ln\left(\frac{z_{ref} - d_0 + z_m}{z_m}\right) + \psi_m\right]^{-1} \tag{39}$$

where $\rho$ is the air density in $\text{kg}/\text{m}^3$, $c_p$ is the specific heat of air at constant pressure in $\text{J}/(\text{kg}{\cdot}°)$, $T_s$ is the temperature of exchange surface in $°\text{C}$, in which the soil surface temperature was considered, $T_a$ is the air temperature at the measured height $z_{ref}$ in $°\text{C}$, in which the temperature at 2 m above the ground was considered, $\gamma_H$ is the resistance to surface

heat transfer (s/m) corrected for atmospheric stability, $\kappa$ is the Karman's constant with a value of 0.41, $u_*$ is the friction velocity (m/s), which is calculated using Equation (50), $d_0$ is the zero plane displacement expressed in m, which is 0.77 times the height of the crop, $z_h$ and $z_m$ are the surface roughness parameters for the temperature and momentum profiles expressed in m, and $\psi_h$ and $\psi_m$ are diabatic correction factors for heat and momentum, respectively.

The calculation of $z_h$ and $z_m$ was related to crop height $h_c$, and $z_m = 0.13h_c$ and $z_h = 0.2z_m$; the determination of $\psi_h$ and $\psi_m$ involved the calculation of atmospheric stability as follows (Campbell, 1977) [51]:

$$s = \frac{\kappa z_{ref} g H}{\rho c_p T_s u_*{}^3} \tag{40}$$

Under stable conditions ($s > 0$),

$$\psi_m = \psi_h = 4.7s \tag{41}$$

Under unstable conditions ($s < 0$) (Norman, 1979) [52],

$$\psi_h = -2\ln\left(\frac{1 + \sqrt{1 - 16s}}{2}\right), \psi_m = 0.6\psi_h \tag{42}$$

$\lambda E_s$ was calculated using the following formula:

$$\lambda E_s = \lambda \rho_w E_s \tag{43}$$

$$\lambda = 2.501 - \left(2.361 \times 10^{-3}\right) T \tag{44}$$

where $\lambda$ is the latent heat in MJ/kg, $\rho_w$ is the density of water in kg/m$^3$, and $E_s$ is the surface evaporation intensity in mm/d.

Determination of Water and Heat Boundary in Water Storage Pit

(1)　Boundary of soil moisture in the pit

Water storage pit irrigation is a variable head infiltration process. With the passage of time, the water level in the pit decreases continuously with the increase of the infiltration amount, thus changing the boundary conditions of the pit wall as well. According to the principle of mass conservation, the process of changing the water level in the pit was processed [17], and the following expression was obtained:

$$H_t = \frac{Vol - \int_0^t \sum_n \left| \sum_{m=1}^N F_{nm} \frac{\theta_m^{j0+1} - \theta_m^{j0}}{\Delta t} + \sum_{m=1}^N A_{Wnm}^{j0+1} h_m^{j0+1} + B_n^{j0+1} + D_n^{j0} \right| dt}{\pi r_0^2} \tag{45}$$

where $H_t$ is the water depth in the pit at moment $t$ in cm, $Vol$ is the amount of irrigation water in a single pit in cm$^3$, $N$ is the total number of nodes in the computing region, and $n$ is the node on the boundary of the pit wall. Additional parameters are detailed in Supplementary Materials.

Equation (45) can be used to obtain the process of changing the water level in the pit over time. Therefore, the boundary conditions can be judged according to the position relationship between $H_t$ and the boundary points in the pit. When the point is lower than $H_t$, the point belongs to the Dirichlet boundary. When the boundary point is higher than $H_t$, the boundary point belongs to the Neumann boundary.

(2)　Boundary of soil temperature in the pit

In comparison with traditional surface irrigation, the water storage pit increases the interface of heat exchange and changes the middle-deep soil temperature. Therefore, the soil hydrothermal simulation of water storage pit irrigation must properly reflect this change.

Under water storage pit irrigation, the radiation in the pit enters from the pit mouth, that is, the total radiation entering the pit is controlled by the pit mouth area, while the heat exchange interface is the pit bottom and the pit wall. Considering the small space in the pit, the calculation can be simplified by assuming that the radiation reaching the pit wall and the pit bottom per unit area is the same, and the total amount is equal to the radiation that enters the pit mouth. The radiation at the pit bottom and pit wall is calculated by the surface radiation at the pit mouth multiplied by the pit coefficient, and the pit coefficient ($f$) can be expressed as follows:

$$f = \frac{S_{kk}}{S_k} \tag{46}$$

where $S_{kk}$ is the pit mouth area in cm$^2$, and $S_k$ is the area of pit bottom and pit wall in cm$^2$.

Under water storage pit irrigation, the net radiation and heat flux density functions of the inner surface of the orchard water storage pit are as follows:

$$R_k = f \cdot R_s \tag{47}$$

$$g_k(t) = R_k - H - \lambda E_s \tag{48}$$

where $R_k$ is the net radiation on the soil surface in the pit in $J/(cm^2 \cdot min)$.

2.3.5. Parameters Determination

The soil in the experimental area of the Fruit Research Institute of Shanxi Academy of Agricultural Sciences was divided into the five following layers: 0–40, 40–70, 70–120, 120–170, and 170–200 cm [17]. The soil hydraulic parameters of each layer are shown in Table 3.

**Table 3.** Soil hydraulic parameters of experimental plot soil.

| Soil Depth/cm | $K_s$/cm·h$^{-1}$ | $\theta_r$/cm$^3$·cm$^{-3}$ | $\theta_s$/cm$^3$·cm$^{-3}$ | $\alpha$ | $n$ |
|---|---|---|---|---|---|
| 0–40 | 0.522 | 0.0545 | 0.51 | 0.0071 | 1.5801 |
| 40–70 | 0.768 | 0.0536 | 0.52 | 0.0080 | 1.5552 |
| 70–120 | 0.684 | 0.0496 | 0.49 | 0.0085 | 1.5296 |
| 120–170 | 0.606 | 0.0465 | 0.50 | 0.0094 | 1.5243 |
| 170–200 | 0.888 | 0.0443 | 0.52 | 0.0091 | 1.5419 |

The fitting parameters of the root water uptake obtained are shown in Table 4 [17].

**Table 4.** Parameters of the root water uptake model.

| $p_x$ | $p_y$ | $p_z$ | $x^*$ | $y^*$ | $z^*$ |
|---|---|---|---|---|---|
| 1.81 | 1.02 | 1.82 | 103.48 | 52.38 | 82.23 |

Chung et al. (1987) [48] determined the $K_h$ and obtained the regression parameters $b_1$, $b_2$, and $b_3$ with values of 0.243, 0.393, and 1.534, respectively.

The evaporation intensity is related to the soil moisture content. The calculation formula for the surface evaporation intensity and pit wall evaporation intensity of water storage pit irrigation orchard obtained by Gu [53] was as follows:

$$e_s = 0.185 \times e^{7.3518\theta} \tag{49}$$

$$e_p = 1.171 \times e^{-76.455|\theta - 0.3261|} \tag{50}$$

where $e_s$ and $e_p$ are the surface and the pit wall evaporation intensity, respectively, expressed in mm/d.

2.3.6. Model Evaluation Indices

To evaluate the model prediction results, we used the root mean square error (RMSE), mean absolute percentage error (MAPE), and minute absolute deviation (MAD) to evaluate the prediction accuracy of the model, and the calculation formulas are as follows:

$$\text{RMSE} = \sqrt{\sum_{i=1}^{l} \frac{\left(\varphi_i^S - \varphi_i^R\right)^2}{l}} \tag{51}$$

$$\text{MAPE} = \frac{1}{l} \sum_{i=1}^{l} \left| \frac{\varphi_i^S - \varphi_i^R}{\varphi_i^R} \right| \times 100\% \tag{52}$$

$$\text{MAD} = \frac{1}{l} \sum_{i=1}^{l} \left| \varphi_i^S - \varphi_i^R \right| \tag{53}$$

where $\varphi$ is the unknown, $\varphi^S$ is the predicted value, $\varphi^R$ is the measured value, and $l$ is the number of points of the measured unknown.

2.3.7. Programming

According to the governing equation and the definite conditions, the calculation program was compiled using MATLAB, defined irrigation at $t = 0$. In the water movement equation, the moisture content $\theta$ of each node was solved and substituted into the heat transfer Equation, and the linear equation system was solved to obtain the corresponding soil temperature $T$.

## 3. Results and Discussions

Based on the field measurement data, the model was verified with the data of 17 days between two irrigation cycles from 24 June to 11 July 2019 (irrigation was carried out on 24 June), and the verified coupling model was used for simulation and analysis.

### 3.1. Model Validation

3.1.1. Comparison and Analysis of Simulated and Measured Values

Three soil moisture monitoring points (1, 2, and 6) were selected, and the simulated values of soil moisture content 1 and 17 days after irrigation were compared with the measured values, as shown in Figure 7. The results showed that at each point, the simulated soil moisture content agreed well with the measured soil moisture content and showed the same variation trend with the change of soil depth. In different time periods after irrigation, the value showed change in terms of water redistribution, the moisture content in the upper soil layer (0–100 cm) gradually decreased after irrigation, and the soil moisture content in the lower soil layer (100–200 cm) did not change much. Finally, after 17 days of irrigation at these three different points, the water content at different depths was approximately 0.2 cm³·cm⁻³, and the distribution tended to be uniform. Therefore, the model can be a good simulation of soil moisture movement in the field.

Four soil temperature monitoring points (7, 8, 9, and 10) were selected, and the simulated values at 1 and 17 days after irrigation were compared with the measured values, as shown in Figure 8. The results showed that the simulated and measured values of soil temperature had the same trend under the conditions of depth change and time change, and the model can be used to study the characteristics of soil temperature change. In comparison with soil water transport, soil temperature had a small range of change, and the temperature of the upper soil was slightly higher than that of the lower soil in each time period. This finding was obtained, because during the test the outside atmospheric temperature was high, the radiation was strong, and the upper soil was greatly affected by the environment, resulting in a relatively high surface soil temperature.

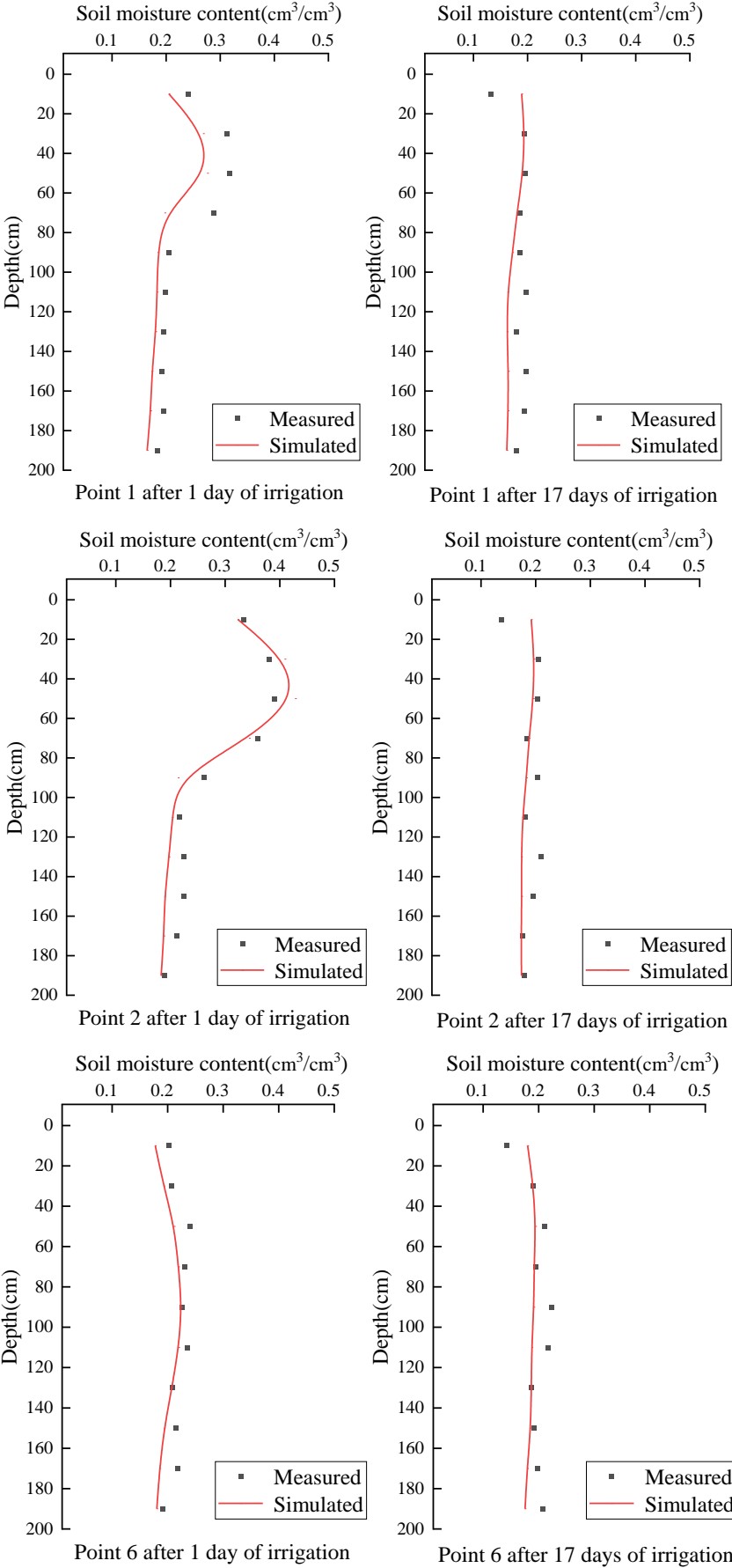

**Figure 7.** Measured and simulated values of soil moisture content at different times and profiles.

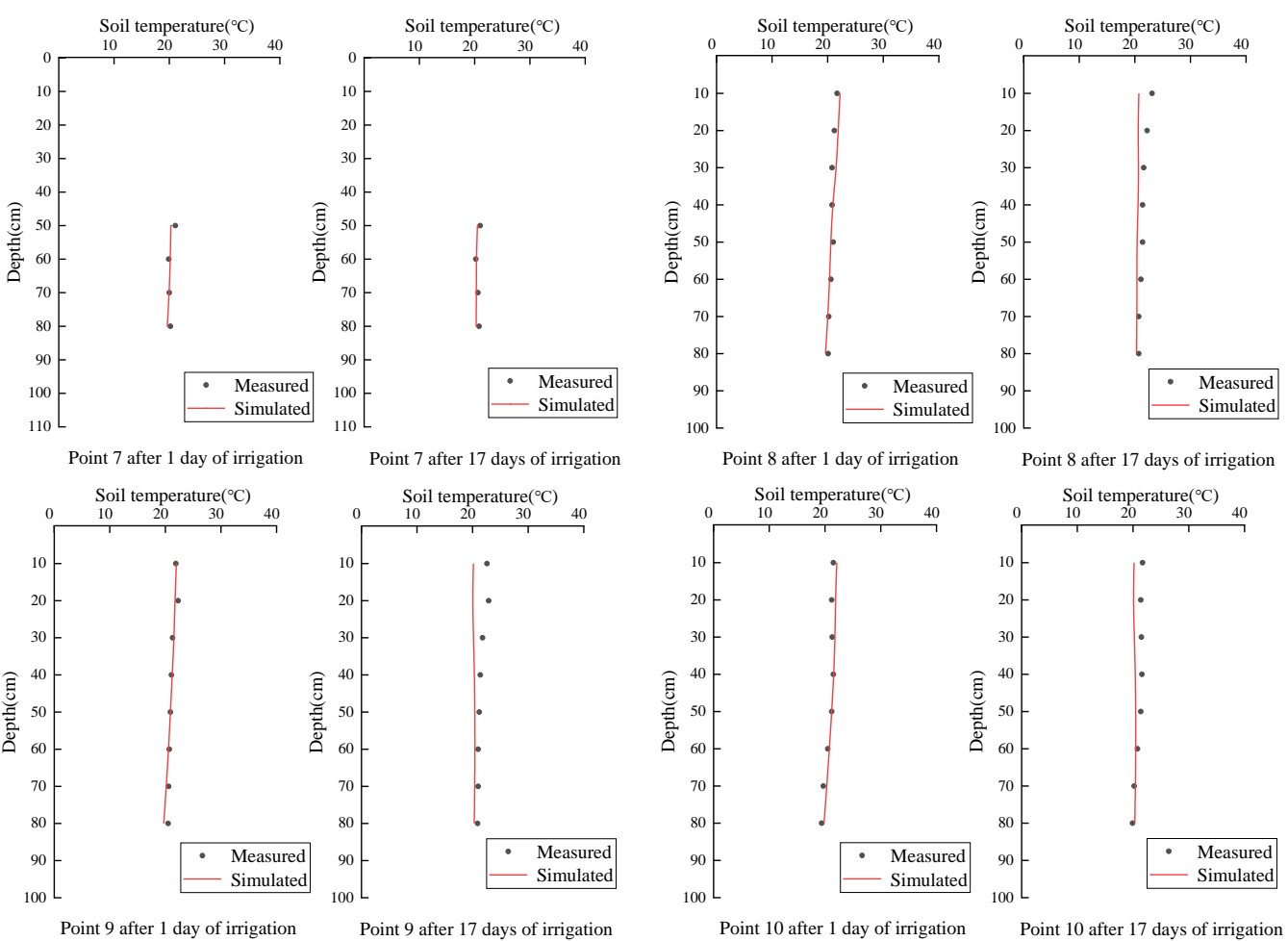

**Figure 8.** Measured and simulated values of soil temperature at different times and profiles.

### 3.1.2. Model Performance Evaluation

The RMSE, MAPE, and MAD indicators of the model were calculated, and the results are shown in Table 5. The RMSE, MAPE, and MAD of the model simulated moisture content were 0.0269, 10.05%, and 0.0214, and the those of the model simulated soil temperature were 0.9460, 3.23%, and 0.6984, respectively. It shows that the soil hydrothermal coupling model for water storage pit irrigation established in this paper has high accuracy in simulating soil moisture and soil temperature.

**Table 5.** Model performance evaluation calculation table.

|  | RMSE | MAPE | MAD |
| --- | --- | --- | --- |
| Soil moisture content | 0.0269 | 10.05% | 0.0214 |
| Soil temperature | 0.9460 | 3.23% | 0.6984 |

### 3.2. *Simulated Interday Dynamic Changes of Soil Water and Heat*

Figure 9 illustrates the distribution of soil moisture content at 1, 5, 11, and 17 days after the model simulated irrigation at an irrigation amount of 300 L and pit depth of 40 cm. The figure shows that 1 day after irrigation, the soil moisture was obviously concentrated around the water storage pit within a range of 80 cm in the horizontal direction and 90 cm in the vertical direction, and presented an elliptic distribution, in which the highest moisture content at the bottom of the pit reached 0.4655 $cm^3 \cdot cm^{-3}$, which spread around. With the passage of time, the range of soil moisture distribution gradually expanded and spread downward and outward from the water storage pit. At 17 days after irrigation,

the water was uniformly distributed in the calculated area, and the coefficient of variation changed from 0.3636 at 1 day after irrigation to 0.0920. The maximum water content gradually decreased with time and was only 0.1990 cm$^3$·cm$^{-3}$ after 17 days of irrigation. The calculation results of statistical indexes are shown in Table 6. The trend is consistent with the findings of Guo et al. (2019) [17]. This phenomenon occurred, because of the water redistribution after irrigation and the reduced moisture content in soil because of the presence of root water absorption and surface evaporation. The results show that the model simulation fully shows the process of water redistribution within 17 days of irrigation, and the model can well simulate the water changes in the orchard field. In comparison with the improvement of soil surface water content by drip irrigation and infiltration irrigation [54], water storage pit irrigation can improve the soil moisture content in the middle-deep layers, which can induce deep rooting of fruit trees and increase the root system in the middle and deep soil [55].

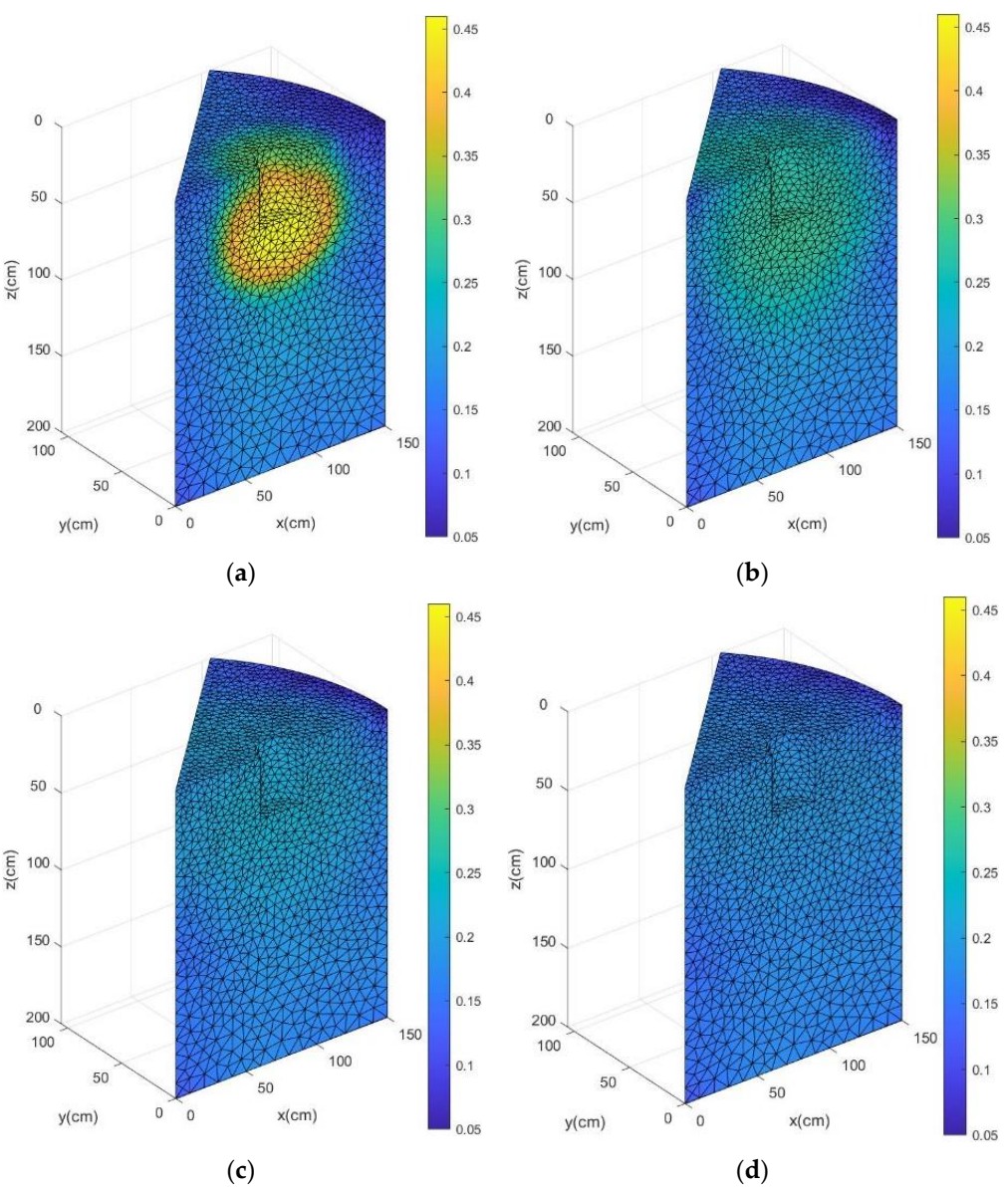

**Figure 9.** Simulation of soil moisture content distribution on different days. (**a**) Soil moisture content after 1 day of irrigation; (**b**) Soil moisture content after 5 days of irrigation; (**c**) Soil moisture content after 11 days of irrigation; (**d**) Soil moisture content after 17 days of irrigation.

**Table 6.** Calculation of statistical indexes of daily soil moisture content for water storage pit irrigation.

| Parameters<br>Date | Average<br>$(cm^3 \cdot cm^{-3})$ | Standard Deviation<br>$(cm^3 \cdot cm^{-3})$ | Coefficient of<br>Variation | Maximum<br>$(cm^3 \cdot cm^{-3})$ | Minimum<br>$(cm^3 \cdot cm^{-3})$ |
|---|---|---|---|---|---|
| 1 day | 0.2166 | 0.0788 | 0.3636 | 0.4655 | 0.0625 |
| 5 days | 0.2088 | 0.0429 | 0.2056 | 0.2822 | 0.0538 |
| 11 days | 0.1924 | 0.0253 | 0.1315 | 0.2279 | 0.0631 |
| 17 days | 0.1795 | 0.0165 | 0.0920 | 0.1990 | 0.0799 |

Figure 10 illustrates the soil temperature distribution at 12:00 at 1, 5, 11, and 17 days after the model simulated irrigation under an irrigation amount of 300 L and pit depth of 40 cm. The figure shows that the daily soil temperature was the highest at the surface and pit wall. With the increase of the distance from the surface and pit wall, the soil temperature gradually decreased. A relatively small high-temperature zone was observed around the heat transfer interface of the surface and pit wall, which was 4–5 cm, and this part also changed dramatically with time. This finding was obtained, because the existence of water storage pit increases the heat exchange interface in middle-deep soil layers, and the surface and the new pit wall interface are both affected by external solar radiation and atmospheric temperature, causing the temperature to change. In addition, the atmospheric temperature was high at 12:00 every day, and the solar radiation absorbed at the surface and the pit wall is high. With the progress of heat conduction, the farther away from the surface and the pit wall, the lower the soil temperature. The maximum soil temperature varied at different times after irrigation, the maximum soil temperatures at 1, 5, 11, and 17 days after irrigation at 12:00 were 24.53, 25.95, 28.32, and 22.98 °C, respectively. The soil temperature on the surface and pit wall reached the highest at 11 days after irrigation and the lowest at 17 days after irrigation. The calculation results of statistical indexes are shown in Table 7. This result was consistent with the atmospheric temperature change during the experiment, indicating that the surface and the pit wall serve as the heat exchange interface, and the soil temperature changes greatly with the change in environmental temperature. The model accurately simulated the soil temperature of water storage pit irrigation under the influence of external environment, such as solar radiation and atmospheric temperature, and reflected the influence of water storage pit on soil temperature, which accurately simulated the distribution characteristics of soil temperature in the field. In comparison with other irrigation methods, the water storage pit irrigation method increases the interface of heat exchange. Therefore, the change of soil temperature near the water storage pit is close to the surface and changes the soil temperature in the middle and deep layers, and in terms of the overall performance, the soil layer farther away from the surface and the pit wall, the lower the soil temperature. The maximum temperature of surface irrigation, sprinkler irrigation, and drip irrigation occurred at the surface, at 5 and 20 cm depth below the surface, respectively, and the further down the soil layer, the lower the temperature [9]. Soil temperature affects the root growth of plant. Yin, et al. (2017) [56] found that the temperature ranging from 18 °C to 26 °C was more conducive to the growth of new roots and shortened the rooting time. When the average daily soil temperature was lower than $17 \pm 0.5$ °C, the root system of birch stopped growing [57]. The model simulated that the lowest soil temperature under water storage pit irrigation was approximately 17 °C, and the overall temperature was in the range of 17–28 °C, indicating that the soil temperature under water storage pit irrigation can result in high growth rate in the root system of the fruit tree.

**Table 7.** Calculation of the statistical indexes of daily soil temperature for water storage pit irrigation.

| Parameters<br>Date | Average<br>(°C) | Standard Deviation<br>(°C) | Coefficient of<br>Variation | Maximum<br>(°C) |
|---|---|---|---|---|
| 1 day | 20.56 | 2.03 | 0.0988 | 24.53 |
| 5 days | 20.68 | 1.85 | 0.0895 | 25.95 |
| 11 days | 21.93 | 2.59 | 0.1183 | 28.32 |
| 17 days | 20.12 | 1.02 | 0.0507 | 22.98 |

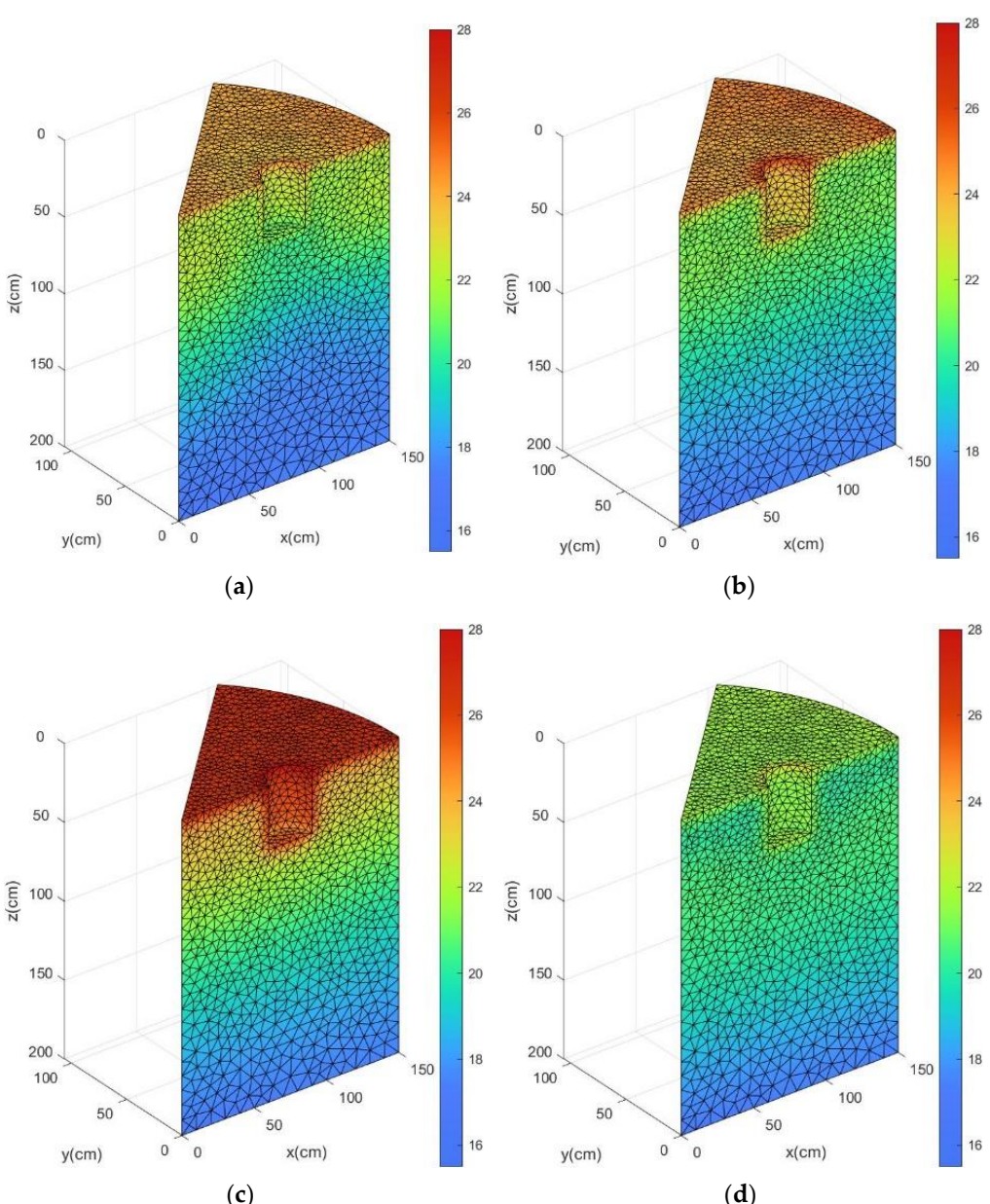

**Figure 10.** Simulation of soil temperature distribution on different days. (**a**) Soil temperature after 1 day of irrigation; (**b**) Soil temperature after 5 days of irrigation; (**c**) Soil temperature after 11 days of irrigation; (**d**) Soil temperature after 17 days of irrigation.

### 3.3. Simulated Intraday Dynamic Changes of Soil Water and Heat

Figure 11 illustrates the distribution of soil moisture content at different times after 1 day of model simulation irrigation at an irrigation amount of 300 L and pit depth of 40 cm. The figure shows that the soil moisture content at 1 day after irrigation was centered on the water storage pit and presented ellipsoidal distribution, and the farther away from the bottom of the pit, the lower the soil moisture content. With the change of time, the highest value of water content gradually decreased, which were 0.5000, 0.4655, 0.3997, and 0.3745 (cm$^3 \cdot$cm$^{-3}$), and the area with high moisture content gradually spread outward. In this condition, as the water redistribution proceeded, the water content distribution gradually became uniform. The calculation results of statistical indexes are shown in Table 8.

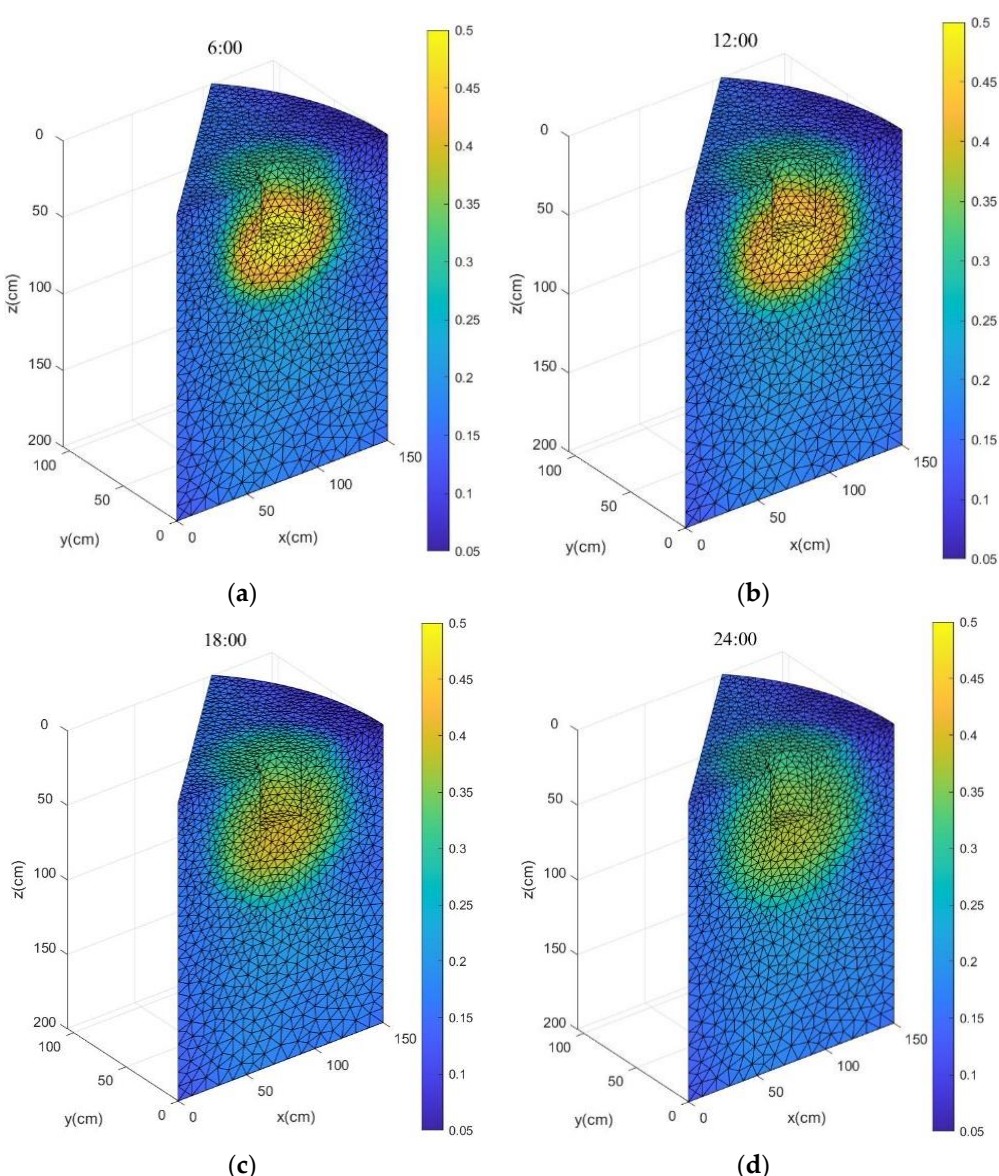

**Figure 11.** Simulation of soil moisture content distribution at different times after 1 day of irrigation. (**a**) 6:00 soil moisture content; (**b**) 12:00 soil moisture content; (**c**) 18:00 soil moisture content; (**d**) 24:00 soil moisture content.

**Table 8.** Calculation of statistical indexes of soil moisture content for water storage pit irrigation day.

| Parameters<br>Time | Average<br>$(cm^3 \cdot cm^{-3})$ | Standard Deviation<br>$(cm^3 \cdot cm^{-3})$ | Coefficient of<br>Variation | Maximum<br>$(cm^3 \cdot cm^{-3})$ | Minimum<br>$(cm^3 \cdot cm^{-3})$ |
|---|---|---|---|---|---|
| 6:00 | 0.2129 | 0.0780 | 0.3664 | 0.5000 | 0.0648 |
| 12:00 | 0.2166 | 0.0788 | 0.3636 | 0.4655 | 0.0626 |
| 18:00 | 0.2144 | 0.0692 | 0.3229 | 0.3997 | 0.0604 |
| 24:00 | 0.2141 | 0.0646 | 0.3016 | 0.3745 | 0.0592 |

Figure 12 illustrates the distribution of soil temperature at different times after 1 day of model simulation irrigation at an irrigation amount of 300 L and pit depth of 40 cm. The figure shows that at 4–5 cm around the pit and below the surface, the soil temperature changed drastically with time, and the intraday temperature in other areas changed slightly. In terms of the variation trend of soil temperature with time, the surface temperature was low at 6:00, the surface temperature rose and remarkably increased the pit wall temperature at 12:00, the surface temperature reached the highest value at four moments

and the temperature high value area extended downwards at 18:00, and the surface and pit wall temperature dropped at 24:00. The calculation results of statistical indexes are shown in Table 9. This trend is consistent with the study of Liao et al. (2021) [8] on the intraday change of soil temperature. The results showed a certain lag in the changes in soil temperature around the surface and pit compared with the changes in atmospheric temperature. The underground temperature was always low, but the temperature of the soil layer, where the water storage pit was located, was always relatively high, indicating that the water storage pit irrigation can keep the suitable temperature of crop root area.

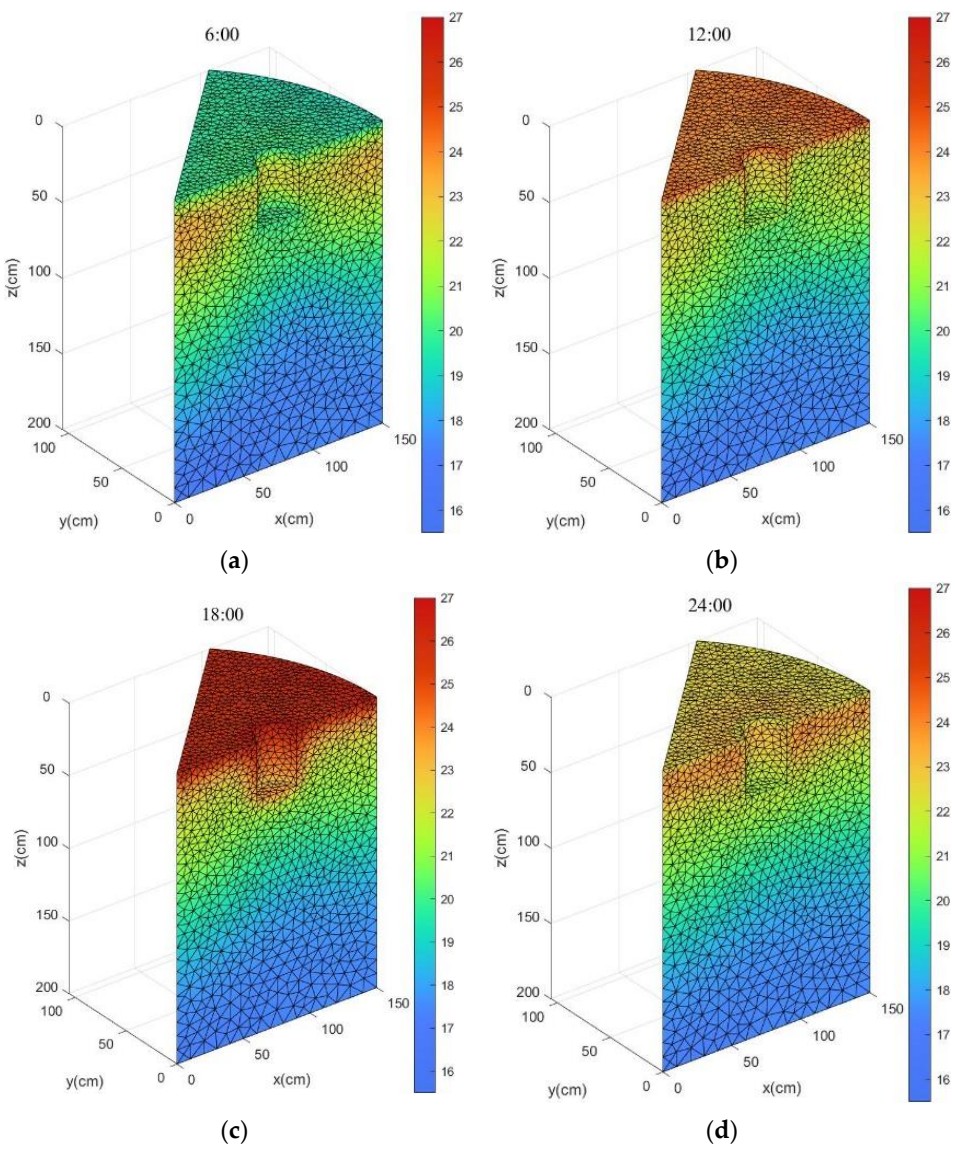

**Figure 12.** Simulation of soil temperature distribution at different times after 1 day of irrigation. (**a**) 6:00 soil temperature; (**b**) 12:00 soil temperature; (**c**) 18:00 soil temperature; (**d**) 24:00 soil temperature.

**Table 9.** Calculation of the statistical indexes of soil temperature for water storage pit irrigation day.

| Parameters Time | Average (°C) | Standard Deviation (°C) | Coefficient of Variation | Maximum (°C) |
|---|---|---|---|---|
| 6:00 | 20.12 | 1.75 | 0.0870 | 23.22 |
| 12:00 | 20.56 | 2.03 | 0.0988 | 24.53 |
| 18:00 | 21.10 | 2.65 | 0.1258 | 26.65 |
| 24:00 | 20.72 | 2.05 | 0.0989 | 23.78 |

*3.4. Simulated the Characteristics of Soil Water and Heat Transfer in Orchard under Different Irrigation Amount*

Figure 13 illustrates the simulation results of soil moisture content distribution after 1 day of irrigation with 40 cm pit depth under different irrigation amounts. Under different irrigation amounts, soil moisture content centers on the water storage pit with an ellipsoid distribution, and the farther away from the bottom of the pit, the smaller the moisture content. With the increase of irrigation amount, the distribution range of high-water-value region expanded, and the values of the highest moisture content also increased, which were 0.3693, 0.4655, and 0.5000 $cm^3 \cdot cm^{-3}$. The calculation results of statistical indexes are shown in Table 10. Combined with the actual fruit tree planting [58], the soil moisture status can be changed by adjusting the amount of irrigation, and the root water absorption and crop growth can be promoted.

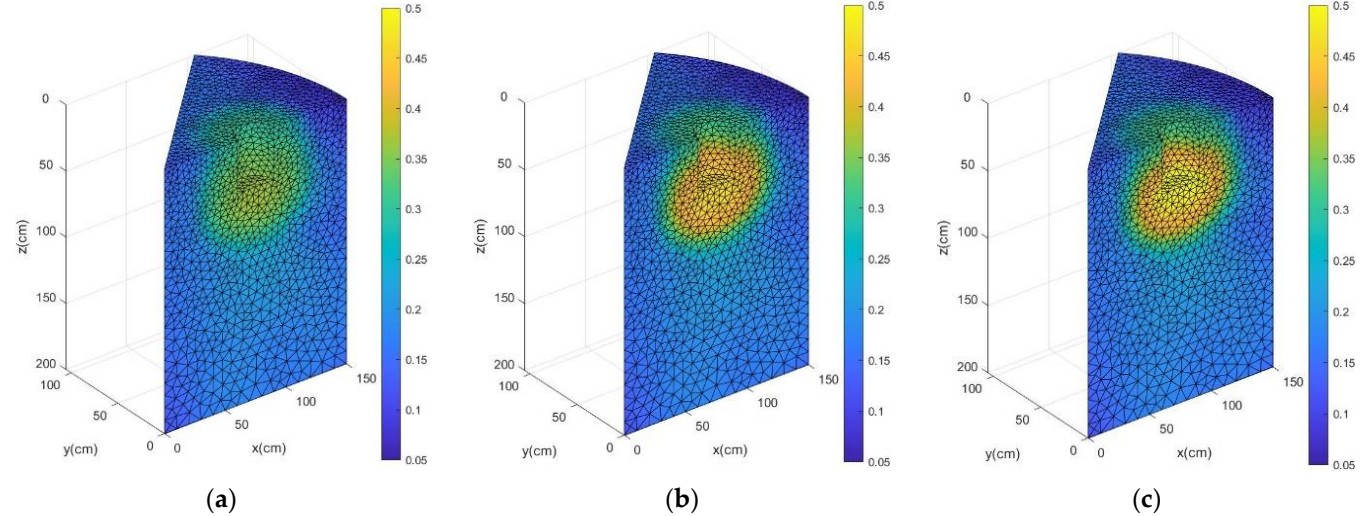

**Figure 13.** Simulation of soil moisture content under different irrigation amounts. (**a**) Irrigation amount: 200 L; (**b**) Irrigation amount: 300 L; (**c**) Irrigation amount: 400 L.

**Table 10.** Calculation of statistical indexes for soil moisture under different irrigation amount.

| Parameters<br>Irrigation Amount | Average<br>($cm^3 \cdot cm^{-3}$) | Standard Deviation<br>($cm^3 \cdot cm^{-3}$) | Coefficient of<br>Variation | Maximum<br>($cm^3 \cdot cm^{-3}$) |
|---|---|---|---|---|
| 200 L | 0.2062 | 0.0586 | 0.2842 | 0.3693 |
| 300 L | 0.2166 | 0.0788 | 0.3636 | 0.4655 |
| 400 L | 0.2177 | 0.0816 | 0.3748 | 0.5000 |

Figure 14 illustrates the simulation results of soil temperature distribution at 12:00 after 1 day of irrigation with 40 cm pit depth under different irrigation amounts. The figures show that under different irrigation amounts, the soil temperature near the surface and pit wall was still higher than the underground temperature except that the soil temperature at the bottom of the pit was low with irrigation of 400 L. The reason is that the soil saturated hydraulic conductivity of the 0–40 cm soil layer was low. When the irrigation amount was 400 L, the water amount was large and the infiltration didn't end at the simulation time. The influence of the irrigation water temperature still existed and the soil temperature at the bottom of the pit was equal to the low irrigation water temperature. With the increase of irrigation amount, the average values of soil temperature were 20.58, 20.56, and 20.52 °C, with a trend of decreasing. The maximum value of soil temperature appeared at the center of the boundary between the water storage pit and the surface and showed the same change trend. The calculation of statistical indexes is shown in Table 11. The reason is that the irrigation water temperature was lower than the middle and upper soil temperature, and the infiltration of irrigation water would lead to the maximum and average value

of soil temperature decrease. With the increase of irrigation amount, the soil minimum temperature gradually increased, which were 17.03 17.19, and 17.20 °C. This finding was obtained, because the lower layer soil temperature is low and receives heat conduction from inside the soil. With the increase of heat absorbed from irrigation and outside, the minimum soil temperature increased. Furthermore, the heat capacity of water was large, and as the soil moisture content increased, the soil heat capacity increased, and the variation range of soil minimum temperature decreased.

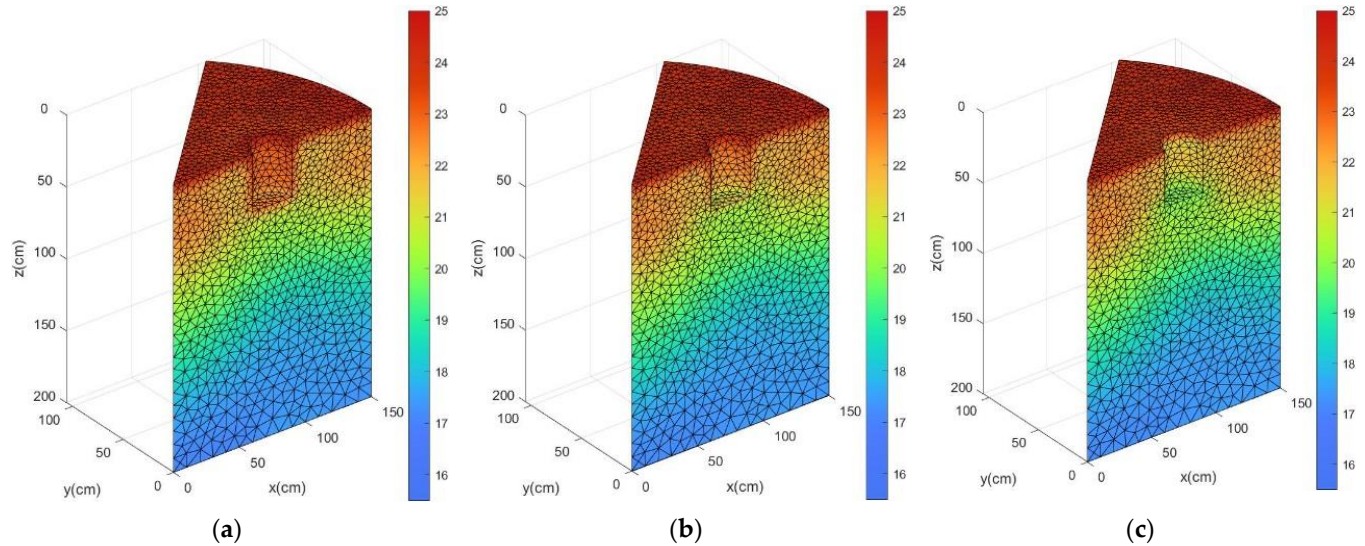

**Figure 14.** Simulation of soil temperature under different irrigation amounts. (**a**) Irrigation amount: 200 L; (**b**) Irrigation amount: 300 L; (**c**) Irrigation amount: 400 L.

**Table 11.** Calculation of statistical indexes for soil temperature under different irrigation amount.

| Parameters / Irrigation Amount | Average (°C) | Standard Deviation (°C) | Coefficient of Variation | Maximum (°C) | Minimum (°C) |
|---|---|---|---|---|---|
| 200 L | 20.58 | 2.04 | 0.0990 | 24.65 | 17.03 |
| 300 L | 20.56 | 2.02 | 0.0988 | 24.53 | 17.19 |
| 400 L | 20.52 | 2.00 | 0.0976 | 24.63 | 17.20 |

*3.5. Simulated the Characteristics of Soil Heat Transfer in Orchard under Different Irrigation Water Temperature*

Figure 15 illustrates the simulation results of soil temperature distribution at 6 h after irrigation at irrigation water temperatures of 15, 20, and 25 °C and pit depth of 40 cm. The figure shows that under different irrigation water temperature conditions, the surface soil temperature was basically unchanged, and in the ellipsoidal area around the pit with high soil moisture content, the soil temperature changed remarkably and violently. With the increase of irrigation water temperature, the temperature of the upper soil (0–70 cm) remarkably increased, and the lowest values at the measurement points were 17.23, 18.89, and 18.95 °C. For every 5 °C increase in water temperature, the soil temperature of the 0–80 cm soil layer increased by 0.003–3.173 °C, and the high-value region of temperature moved down. The calculation of statistical indexes is shown in Table 12. The temperature of the lower soil (70–200 cm) did not change much, because the surface soil temperature is mainly affected by solar radiation, while the soil temperature of pit wall and bottom is mainly affected by water temperature in a short period (18–24 h) after irrigation. However, considering the limitation of water infiltration distance, the range of influence of water temperature is limited. When the irrigation water temperature is very low, the soil temperature is low, which is not conducive to crop growth (Rashid et al., 2019) [59]. Therefore, when using groundwater or low-temperature water for irrigation,

the water transport distance can be extended or biochar can be applied to improve the surface soil temperature (Ding, et al., 2019) [10].

**Figure 15.** Simulation of soil temperature under different irrigation water temperatures. (**a**) Water temperature: 15 °C; (**b**) Water temperature: 20 °C; (**c**) Water temperature: 25 °C.

**Table 12.** Calculation of statistical indexes for 0–80 cm soil temperature under different irrigation water temperature.

| Parameters Irrigation Water Temperature | Average (°C) | Standard Deviation (°C) | Coefficient of Variation | Maximum (°C) | Minimum (°C) |
|---|---|---|---|---|---|
| 15 °C | 20.89 | 1.95 | 0.0932 | 25.27 | 17.23 |
| 20 °C | 21.61 | 1.74 | 0.0807 | 25.36 | 18.89 |
| 25 °C | 22.33 | 1.96 | 0.0878 | 25.45 | 18.95 |

## 4. Conclusions

In this paper, considering the effects of root water uptake, precipitation, evaporation, irrigation, canopy, and water storage pits, a three-dimensional soil hydrothermal coupling model of orchard under water storage pit irrigation was established. Galerkin finite element method was used to solve the model, and field measurement data were used to verify and analyze the simulation results. Results showed that the model has high accuracy in simulating soil moisture movement and heat transfer and can be used to simulate soil moisture and heat transfer in water storage pit irrigation.

The simulation results showed that the soil moisture content centers on the water storage pit with an ellipsoid distribution and tends to be uniform over time. The soil temperature was the highest at the soil surface and pit wall, and the temperature at 4–5 cm changed dramatically with time. The farther away from the soil surface and pit wall, the lower the soil temperature. The intraday variation of soil temperature in the area around the pit and surface was severe, and a lag was observed compared with the variation of atmospheric temperature. Under different irrigation amount conditions, with the increase of irrigation water, the distribution range of the high-water-value region and the maximum moisture content increased, whereas the average and maximum soil temperature decreased, and the minimum soil temperature gradually increased. Under different irrigation water temperature conditions, with the increase of irrigation water temperature, the soil temperature remarkably increased in the ellipsoidal area around the pit, and the temperature in the 0–80 cm layer increased by 0.0006–0.6346 °C·°C$^{-1}$, but the influence of water temperature only lasted for 18–24 h after irrigation.

Current research shows that the water storage pit irrigation method can effectively improve the soil moisture content in the middle and deep layers, improve the temperature

of the middle and deep soils, and then induce the root system of the fruit tree to be deeply rooted. It can also promote the water absorption of the root system and enhance the growth of the fruit tree. The model established in this paper can simulate the soil water and heat conditions under the complex conditions of the water storage pit irrigation, and can be used to guide the management of water resources in the field. From the simulation results, different irrigation amounts and irrigation water temperatures will affect the water and heat conditions in the soil, which in turn affects the growth of fruit trees. In order to seek the optimal combination of conditions, maximize the beneficial coupling of soil water and heat, increase fruit tree yield and improve fruit quality, we need to further explore, link the root growth status of fruit trees and the yield quality of fruit trees with the soil hydrothermal condition under water storage pit irrigation, and intuitively guide the actual agricultural production.

**Supplementary Materials:** The following supporting information can be downloaded at: https://www.mdpi.com/article/10.3390/w14111813/s1, Figure S1: Tetrahedral dissection results; Details of Tetrahedral element Galerkin finite element equation discretization and Programming. Reference [60] is cited in the supplementary materials.

**Author Contributions:** Conceptualization, X.S.; investigation, L.Z., L.H. and F.H.; software, X.G.; writing—original draft, Y.S.; writing—review and editing, T.L. and J.M. All authors have read and agreed to the published version of the manuscript.

**Funding:** This research was funded by State Key Laboratory Open Research Fund Project (grant number IWHR-SKL-202110) and Water Conservancy Science and Technology Research and Promotion Project in Shanxi Province (grant number 2022GM012).

**Acknowledgments:** The authors thank the Fruit Research Institute of Shanxi Academy of Agricultural Sciences for providing the site for experiment and collecting data.

**Conflicts of Interest:** The authors declare no conflict of interest.

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
