# Peer review of "Three-Dimensional Model of Soil Water and Heat Transfer in Orchard Root Zone under Water Storage Pit Irrigation"

_water, doi:10.3390/w14111813_

Round 1

Reviewer 1 Report

Please see a attachment

Author Response

请参阅附件。

Reviewer 2 Report

Dear Authros,

This is an interesting paper, but it reads where heavy.

You need to add missing info, but at the same time make it more clear and readable, moving parts to supplementary material.

Please find below my comments to improve it.

Introduction

This  is heavily underrefrenced till line 58. Provide reefrences for your major statements.

Introduce FIg 1 at approx line 80, when you descrbe the pit irrigation method.

M and M

Add a map to locate the study site

move the rainfall data for the experiment to the experimental design section.

line 144-147, so you only measured these variables twice. Please state why.

Provide also a pic for the vertical measurement set up.

Make reference to fig 1 already in lines 150-154

Figure 2. provide a legend.

Show the measured data in graphs or tables.

On eof the main element is that you do not specify the irrigation schedule make a single graph showing T, rainfall and irrigation time (experiment time should be on x axis).

From section 2.32 the paper reads really heavy. Reduce the number of sub-sections (only create subsections for boundary conditions, inital conditions, etc.)

Make it clearer (i.e. when in line 244 you wirte ALDFEI write you refer to fig 5).

Line 324 - provide refernce

Bring all the section form 322 to 343 to supplementary material.

Line 429: maybe is "mean" instead than "minute"?

Results and discussion

if you do not specify how irrigation time worked out - your results are hardly undertsandable.

Is the irrigation run under constant head condition in the pit? for how long?

you need to set it clearly, maybe in  a table/chart form... not hidden in the middle of the text...

Take care of titles. i.e. title for sect 3.2 should read "Simulated interday..."

Conclusion

As they are written now, they are not useful to the purpose of the paper. Please provide comments of broader breath for the results of your work

Author Response

请参阅附件。

Round 2

Reviewer 2 Report

In my opinion the paper is ready for publication

This manuscript is a resubmission of an earlier submission. The following is a list of the peer review reports and author responses from that submission.